# Reducibility of Al^3+^-Modified Co_3_O_4_: Influence of Aluminum Distribution

**DOI:** 10.3390/ma16186216

**Published:** 2023-09-14

**Authors:** Svetlana V. Cherepanova, Egor G. Koemets, Evgeny Yu. Gerasimov, Irina I. Simentsova, Olga A. Bulavchenko

**Affiliations:** 1Boreskov Institute of Catalysis SB RAS, Lavrentiev Ave., 5, Novosibirsk 630090, Russia; 2Department of Physics, Novosibirsk State University, Pirogova, 2, Novosibirsk 630090, Russia

**Keywords:** reducibility, cobalt oxide, in situ XRD, solid solution, catalyst, modification

## Abstract

The reduction of Co-based oxides doped with Al^3+^ ions has been studied using in situ XRD and TPR techniques. Al^3+^-modified Co_3_O_4_ oxides with the Al mole fraction Al/(Co + Al) = 1/6; 1/7.5 were prepared via coprecipitation, with further calcination at 500 and 850 °C. Using XRD and HAADF-STEM combined with EDS element mapping, the Al^3+^ cations were dissolved in the Co_3_O_4_ lattice; however, the cation distribution differed and depended on the calcination temperature. Heating at 500 °C led to the formation of an inhomogeneous (Co,Al)_3_O_4_ solid solution; further treatment at 850 °C provoked the partial decomposition of mixed Co-Al oxides and the formation of particles with an Al-depleted interior and Al-enriched surface. It has been shown that the reduction of cobalt oxide by hydrogen occurs via the following transformations: (Co,Al)_3_O_4_ → (Co,Al)O → Co. Depending on the Al distribution, the course of reduction changes. In the case of the inhomogeneous (Co,Al)_3_O_4_ solid solution, Al stabilizes intermediate Co(II)-Al(III) oxides during reduction. When Al^3+^ ions are predominantly on the surface of the Co_3_O_4_ particles, the intermediate compound consists of Al-depleted and Al-enriched Co(II)-Al(III) oxides, which are reduced independently. Different distributions of elemental Co and Al in mixed oxides simulate different types of the interaction phase in Co_3_O_4_/γ-Al_2_O_3_-supported catalysts. These changes in the reduction properties can significantly affect the state of an active component of the Co-based catalysts.

## 1. Introduction

Co-based catalysts are widely utilized in different catalytic reactions. However, various states of cobalt are active under different conditions of catalytic reactions. Co_3_O_4_ is the most active catalyst among transition metal oxides used for the complete oxidation of CO, hydrocarbons, volatile organic compounds (VOCs), diesel soot, phenols, etc. [1,2,3]. The use of alumina supports increases the lifetime of Co_3_O_4_ catalysts for CO oxidation [4,5]. Catalysts supported on alumina and containing Co_3_O_4_ as the main phase of cobalt and CoO_*x*_ species, due to interactions with alumina, have proven to be active for both water–gas shift (WGS) and methanation. During ethanol reformation, the CO produced can react with water (WGS) or hydrogen (methanation, without water) on Co sites [6,7]. CoO is a well-known photocatalyst for the overall splitting of water into H_2_ and O_2_ [8,9]. In some reactions, the synergetic effect is achieved with the use of CoO-Co composite catalysts [10,11,12,13,14,15]. For water splitting via electrolysis, the catalytic activity of CoO is restricted by its weak conductivity. One of the most well-known ways of inducing the reduction of reaction barriers and improving conductivity is to combine cobalt oxides with highly conductive materials, for example, metallic cobalt. Adjusting the phase proportions of Co and CoO, the charge transfer process between Co and CoO appears to show impressive electrocatalytic performance [10,11]. Aligned Co@CoO_*x*_ nanostructures efficiently electrocatalyze the oxygen evolution reaction (OER), as the CoOx layer provides the necessary active catalytic sites while the Co metal improves the electronic conductivity, as compared with insulating or semi-conductive CoO_*x*_-based materials [12]. Coexisting Co^0^ and CoO phases turn out to be an effective composite catalyst for the selective catalytic transfer hydrogenation of fatty acids to fatty alcohols using biomass-derived isopropanol as a hydrogen donor. Such catalysts were prepared via the calcination of Co-Al layered double hydroxides (LDHs), followed by reduction at different temperatures, and the Co/Co^δ+^ ratio was simply tuned at different reduction temperatures [13]. A similar approach was applied for the fabrication of catalysts for the selective hydrogenation of CO_2_ into ethanol. The catalysts were also derived from Co-Al LDH. The efficiency of CO_2_ hydrogenation to ethanol with these catalysts was optimized by adjusting the Co-CoO composition with different pre-reduction temperatures [14]. In another study, the hydrogenation of CO_2_ to CO and CH_4_ was induced by CoO and Co^0^ sites, respectively, via the different reactive pathways [15]. Therefore, the strategy of tuning the composition of catalysts offers opportunities for the optimization of the effective hydrogenation of CO_2_ to desired products. Metallic cobalt is the most widely accepted active phase in Fischer–Tropsch synthesis (FTS) [16,17,18,19,20,21]. FTS is an industrial catalytic process of the hydrogenation of CO to paraffins, olefins, and oxygenated hydrocarbons. While cobalt mainly yields linear paraffins [22], cobalt oxides often coexist in the catalyst under FTS reaction conditions, as the presence of water promotes the reoxidation of small cobalt nanoparticles [21]. The oxidation of nano-sized metallic cobalt to cobalt oxide during FTS is often considered to be a major deactivation mechanism [19]. However, studies on the deactivation of alumina-supported cobalt catalysts have shown that oxidation is not a deactivation mechanism during realistic FTS, as, in these studies, the environment was found to be strongly reducing [23,24].

Thus, different reactions require different cobalt valence states of cobalt and, consequently, different resistance levels of cobalt oxides to reduction. It is worth noting that Co_3_O_4_ is easily reduced to metallic cobalt via two sequential steps: Co^3+^ →Co^2+^ →Co^0^. This is confirmed by the presence of two narrow peaks in the temperature-programmed reduction (TPR) data, with the maxima in the range of T = 270–415 °C, whilst H_2_ consumption agrees well with the stoichiometry of the following two reduction steps: Co_3_O_4_→CoO→Co [25,26,27,28,29,30]. Various modifiers might impact the reduction process, either facilitating or impeding it, thereby affecting the efficiency of a catalyst. For example, unmodified Co_3_O_4_ is not considered to be a suitable catalyst in WGS reactions due to the low thermal stability of reducible cobalt oxides. However, Mn-modified mesoporous Co_3_O_4_ (15 wt% Mn) shows superior CO conversion and structural stability compared to unmodified mesoporous Co_3_O_4_ [7,31]. In one study, it was shown that Mn ions mainly complicated the second step of reduction [32]. The introduction of Al_2_O_3_ as a support [33] or modifier [34] also inhibits cobalt oxide reduction due to the interaction between cobalt and aluminum oxides. Thus, the lower mobility of the Al_2_O_3_ modifier, caused by its interaction with cobalt particles on the surfaces of the mesoporous Co_3_O_4_, improved the FTS activity and stability, with less coke or wax deposition on the active sites [34]. In some cases, interaction led to the full dissolution of Al^3+^ ions in Co_3_O_4_, along with the formation of (Co,Al)_3_O_4_ solid solutions, which in turn demonstrate catalytic properties in methane decomposition [35], OER [36], toluene total oxidation [37], and formaldehyde production from emissions of methanol and methanethiol [38]. The substitution of Co^2+^ and Co^3+^ ions by Al^3+^ ions modulates the catalysts’ surface morphology towards a more electrocatalytically active one and tailors the energy level for faster charge transfer in (Co,Al)_3_O_4_ samples, which are highly efficient electrocatalysts for water oxidation [36]. The optimized Co_1_._75_Al_1_._25_O_4_ nanosheets exhibit unprecedented OER activity. In the case of CH_4_ decomposition, the state of the initial oxide changes due to activation with H_2_. Catalytic activity was related not to the amount of Co in the catalyst but rather to the size of crystallites on the activated sample [35]. Samples with Co/Al ratios between 1 and 2.3 presented the best activity and stability because they initially consisted of dispersed Co-Al mixed oxides, which produced a dispersed active phase.

While the TPR characterization of Co_3_O_4_, promoted or supported by alumina, has been performed in many studies, there are different interpretations of almost identical TPR profiles in the literature [39]. Moreover, some features of the reduction process are often explained in the concept of a “strong interaction” between Co and Al species [17,40]. In general, the interaction between cobalt oxide and alumina results in the formation of a double compound, a homogeneous or inhomogeneous solid solution, a core–shell structure, and so on. All these cases are usually called a strong interaction, and there is a lack of information on its structural characteristics.

The present work is focused on the study of the reducibility of Co-Al oxides with different distributions of Al^3+^ ions, which simulate different types of the interaction phase in Co_3_O_4_/γ-Al_2_O_3_-supported catalysts during reduction. To this end, coprecipitated Co-Al samples with Al/(Co + Al) = 1/6, 1/7.5 were calcined at 500 and 850 °C, leading to different states of inhomogeneity in mixed oxides. The modification of Co_3_O_4_ and regulation of its reducibility expand the possibilities of catalyst application in different areas. In this work, we use TPR by H_2_ and in situ X-ray Diffraction (XRD) to investigate the behavior of Co-Al oxides during their reduction in H_2_. HAADF-STEM with EDS mapping was applied to study the cation distribution over the particles.

## 2. Materials and Methods

### 2.1. Preparation

Co-Al oxides with the Al fraction x = Al/(Co + Al) = 0, 1/7.5, 1/6 were prepared via the coprecipitation of Co^2+^ and Al^3+^ ions from 10 wt% aqueous solutions of their nitrates with subsequent calcination in air at 500 and 850 °C for 4 h. The following nitrates were used: Co(NO_3_)_2_∙6H_2_O of analytical grade (Ural Factory of Chemical Reagents, Ekaterinburg, Russia) and Al(NO_3_)_3_∙9H_2_O of analytical grade (Panreac, Barcelona, Spain). A sodium carbonate solution was used as the precipitant (Na_2_CO_3_, chemically pure grade, State Standard 83–79). The samples were precipitated at pH 7.1–7.2 and a temperature of 65–68 °C. The obtained precipitate was carefully washed out with distilled water to remove sodium ions and air-dried under an infrared lamp at 45 °C for 24 h. The cationic composition of the synthesized samples was verified via atomic emission spectrometry on an Optima 4300 DV instrument (Perkin Elmer, Waltham, MA, USA). According to atomic emission spectrophotometry, the sodium content in the dried samples did not exceed 0.005% by weight.

### 2.2. X-ray Diffraction

The X-ray diffraction (XRD) patterns of oxides were recorded using an X-ray diffractometer (Thermo, CuKα radiation) by scanning in the 2θ-angle range from 15 to 141° with a step of 0.02° and a rate of 2°/min.

In situ XRD experiments were carried out on a precision X-ray diffractometer mounted in the synchrotron radiation extraction channel of a VEPP-3 electron storage ring (the Siberian Synchrotron and Terahertz Radiation Center). The precision X-ray diffractometer included a monochromator, a collimation system, and a position-sensitive detector. The single-reflection Ge(111) crystal monochromator deflected the monochromatic beam upward by ∼30° in the vertical plane and afforded the monochromation of Δλ/λ ≈ (2–3) × 10^−4^ with a radiation wavelength of λ = 1.731 Å. The diffractometer was equipped with an XRK-900 high-temperature reactor chamber (Anton Paar, Graz, Austria). The sample was loaded in an open holder, which allowed the reducing mixture to penetrate into the bulk of a powder sample placed in the reactor chamber. The chamber was arranged on the diffractometer so that the monochromatic beam of synchrotron radiation was incident to the sample surface at an angle of ∼15°. The calcined samples were heated from room temperature to 400, 500, and 580 °C at a rate of 2 °C/min in a 100% H_2_ flow, with a flow rate of 1.5 mL per sec.

The refinement of lattice constants and structural parameters and the determination of average crystallite sizes were carried out via full-profile Rietveld analysis with the use of TOPAS 4.2 software (Bruker, Karlsruhe, Germany).

### 2.3. Temperature-Programmed Reduction

TPR was performed in a mixture of 10% H_2_/Ar (flow rate 40 cm^3^/min, pressure 1.0 bar) passing over the samples. The samples were heated from room temperature up to 900 °C with a constant temperature ramp rate (10 °C/min). Before the reduction, the samples were treated at 500 °C for 0.5 h and then cooled down to room temperature in the O_2_ flow.

### 2.4. Transmission Electron Microscopy

Transmission electron microscopy (TEM) images were obtained using a ThemisZ (Thermo Fisher Scientific, Waltham, MA, USA) electron microscope with an accelerating voltage of 200 kV and a maximum lattice resolution of 0.07 nm. TEM images were recorded with a Ceta 16 (Thermo Fisher Scientific, Waltham, MA, USA) CCD matrix. High-angle annular dark-field imaging (HAADF STEM image) was performed using a standard ThemisZ detector. Energy-dispersive X-ray spectroscopy (EDS) element mapping was obtained using a SuperX Thermo Fisher Scientific energy dispersive spectrometer. The samples were fixed on standard copper grids using ultrasonic dispersion in ethanol.

## 3. Results

### 3.1. TPR-H_2_

The reducibility of all Co-Al samples in the present work was investigated by TPR-H_2_ (Figure 1). Pure Co_3_O_4_, obtained by calcination at 500 °C, showed two partially overlapped narrow peaks of hydrogen consumption between 200 and 360 °C with maxima at T_1_ ~ 280 °C and T_2_ ~ 325 °C. The ~45 °C difference between the maxima of the TPR peaks is in agreement with those reported in the literature: T_1_ = 300 °C and T_2_ = 350 °C [25] and T_1_ = 319 °C and T_2_ = 367 °C [26]. The first and second peaks correspond to the reduction of Co_3_O_4_ to CoO and CoO to Co, respectively. 

In contrast, the reduction of samples with the Al fraction x = 1/7.5 and 1/6, obtained by calcination at 500 °C and 850 °C, is more complicated compared to Co_3_O_4_. Both Co-Al samples calcined at T = 500 °C show a continuous reduction in the temperature range of 200–750 °C with a slight decrease in hydrogen consumption at T~450–500 °C. Therefore, the TPR profiles can be divided into two very broad TPR peaks with maxima at T_1_ = 300–345 °C and T_2_ = 610–690 °C. Some similarity can be found between these TPR profiles and typical TPR data for γ-alumina-supported Co_3_O_4_. In the Co content range of 15%Co–25%Co, Co_3_O_4_/Al_2_O_3_ generally exhibits two TPR peaks up to 800°, a relatively sharp peak from T~200 to ~400 °C with a maximum at T~350 °C, and a broad peak that extends from T~400 to ~800 °C. There are different interpretations for the appearance of two TPR peaks, according to Jacobs et al. [39], who reported the summarized information for the similar TPR profiles of alumina-supported Co_3_O_4_. Case #1 is a two-step reduction where the first and second TPR peaks are related to reactions Co_3_O_4_ + H_2_→3CoO + H_2_O and CoO + H_2_→Co^0^ + H_2_O, respectively. This interpretation signifies that the second step is heavily dependent on the interaction with alumina. Case #2 and Case #3 can be united by the assumption that the direct fast reduction Co_3_O_4_→Co^0^ of large Co_3_O_4_ particles not interacting with alumina occurs (the first TPR peak). The second TPR peak is then assumed to arise from a complicated reduction of highly dispersed cobalt oxide phases strongly interacting with alumina. In some cases, one very broad peak is observed in the range of T = 300–800 °C for Co_3_O_4_/Al_2_O_3_ in the loading range of 6%Co−20%Co [41].

The samples calcined at 850 °C show three distinct TPR peaks with maxima at T_1_~345–350 °C, T_2_~430–440 °C, and T_3_~600–620 °C. Similar TPR profiles were observed on the inverse model catalysts, which were prepared via the impregnation of Co_3_O_4_ particles with small amounts of alumina in the range of 0.1%Al–2.5%Al [42]. It was shown that the introduction of a small amount of alumina to the Co_3_O_4_ surface decreases its reducibility. With increasing alumina loading and increasing calcination temperature, two TPR peaks of Co_3_O_4_ shift to higher temperatures, and additional peaks above 400 °C appear. Peaks in the ranges of T = 400–650 and 650–800 °C were assigned to the formation of a non-stoichiometric cobalt–alumina phase with cobalt ions in octahedral and tetrahedral coordination, respectively [42].

Excluding standard samples, the interpretation of TPR data might be ambiguous, since TPR is an indirect method that allows for hydrogen consumption to be monitored at different temperatures only. Information on the structure of the species present is less straightforward than, for instance, that obtained by in situ XRD and/or TEM, which are complementary to the TPR method. Therefore, it is desirable to apply them in combination.

### 3.2. Transmission Electron Microscopy

TEM images of Co-Al samples calcined at T = 500 and 850 °C show that the particle size varies from 10 to 25 nm and from 100 to 200 nm, respectively (Figure 2a,b). According to High-Resolution TEM combined with fast Fourier transform (FFT), after synthesis, Co-Al samples correspond to the spinel structure (Figure 2c,d). However, it is not possible to differ between Co_3_O_4_ and (Co,Al)_3_O_4_ solid solution due to the very close lattice constants of Co_1-x_Al_x_O_4_ (x = 0 ÷ 1/5). For example, the values of the lattice constants for a = 8.084 Å (Co_3_O_4,_ x = 0; PDF#43-1003), a = 8.086 Å (Co_2_AlO_4_, x = 1/3; PDF#38-0814), and a = 8.104 Å (CoAl_2_O_4_, x = 2/3; PDF#44-0160) are very close to each other. Consequently, the interplanar distances are almost the same within the error of their determination byHRTEM.

The distribution of Co and Al in the Co-Al oxides prepared via calcination at different temperatures was monitored by using HAADF-STEM combined with EDS element mapping. It was observed that particles in the Co-Al sample with the Al fraction x = 1/6 calcined at T = 500 °C were not completely homogeneous in composition (Figure 3a). However, most of the particles were Al-enriched, and a small part was Al-depleted.

After calcination at T = 850 °C, the particles became larger and more uniform. However, there is an inhomogeneous distribution of Al over the particles’ volume. It seems that a high temperature induces the migration of Al^3+^ ions to the particle surface, leading to the formation of particles with an Al-depleted interior and Al-enriched surface (Figure 3b,c). Such an Al-enriched surface of Co_3_O_4_ particles could arise because of the sintering and decomposition of the initial solid solution (Co,Al)_3_O_4_. This result correlates well with the literature data [5], where surface enrichment by Al^3+^ was detected after calcination at T = 700 °C using X-ray photoelectron spectroscopy. Presumably, a driving force for these surface Al^3+^ enrichments is the reduction of Co^3+^ to Co^2+^ in an oxidizing environment around 700–1000 °C [43].

### 3.3. XRD

The XRD patterns of all of the samples correspond to the cubic spinel structural type (as of Co_3_O_4_, Figure 4). For the samples calcined at 500 °C, the addition of Al^3+^ ions leads to a decrease in the average crystallite size from D = 24 to 12 and 10 nm for x = 1/7.5 and x = 1/6, respectively (Table 1). The lattice constant increases from a = 8.085(1) Å (x = 0) to a = 8.087(1) (x = 1/7.5) and 8.091(1) Å (x = 1/6). Calcination at 850 °C increases the average crystallite size to D = 135 nm for Co_3_O_4_ and to D = 90 nm for both (Co,Al)_3_O_4_ samples. As shown in the following, the lattice constants remain almost the same: a = 8.085 Å (x = 0), a = 8.088 Å (x = 1/7.5), and a = 8.090 Å (x = 1/6) (Table 1). The increased lattice constant for the Al-containing samples can be explained by the larger ionic radii of Al^3+^ ions (r = 0.68 Å, CN = 6) compared to the ones of Co^3+^ (r = 0.55 Å, CN = 6) [44]. Despite the higher accuracy in the determination of interplanar spacings and lattice constants, compared to HRTEM-FFT, the XRD technique cannot differentiate between samples containing particles with uniform (Co_1-x_Al_x_)_3_O_4_ (x = 1/6, 1/7.5) composition and samples containing a mixture of Co-Al oxide particles with different Al contents or Co-Al oxide particles with an Al-enriched surface.

### 3.4. In Situ XRD

To interpret the TPR peaks (Figure 1), we carried out in situ XRD experiments. Samples with the Al fraction x = 1/6 calcined at T = 500 and 850 °C were reduced in situ in a H_2_ flow at 400, 500, and 580 °C.

For the sample calcined at 850 °C, the peaks of Co_3_O_4_ and CoO, as well as the peaks of metallic fcc and hcp cobalt can be seen after reduction at 400 °C (Figure 5). Reduction at 500 °C leads to the disappearance of Co_3_O_4_ reflections and an increase in the intensity of the peaks of metallic cobalt (mainly fcc). Peaks of CoO are shifted to higher diffraction angles, which indicates a decrease in the lattice constant with temperature. At 580 °C, the XRD patterns change towards decreasing the intensity of CoO peaks and increasing that of the metallic fcc cobalt peaks. These observations can be interpreted as follows: after reduction at T = 400 °C, two types of CoO containing relatively low and relatively high concentrations of Al^3+^ ions are formed. This is manifested by the asymmetry of the CoO (200) diffraction peak at 2θ = 47–50°. After reduction at T = 500 °C, the Al-depleted CoO is reduced to Co^0^, and Al-enriched Co_3_O_4_, still existing at 400 °C, is reduced to the Al-enriched CoO with a smaller lattice constant compared to the Al-depleted CoO. A decrease in the lattice constant’s dependence on the Al content was also observed for Al^3+^-modified MgO and NiO, where a higher Al content reduced the lattice constant [45]. This effect is due to the presence of Al^3+^ cations, which are smaller than Co^2+^, Mg^2+^, and Ni^2+^. Moreover, the partial substitution of Co^2+^ by Al^3+^ ions leads to the appearance of cationic vacancies to compensate for the excess positive charge. The substitution of ions by vacancies also decreases the lattice constant due to lattice contraction around vacancies. For example, it was found that the effective radii of the oxygen vacancies (r~0.9–1.1Å) in fluorite-like oxides are significantly smaller than the radius of the oxygen ion O^2−^ (r = 1.38Å) [46]. The second feature of Al-enriched CoO is a peak of diffuse scattering on the left side of the 111 peak of Al-enriched CoO (shown by the vertical arrow in Figure 5). At 580 °C, some part of the Al-enriched CoO is reduced to Co^0^ (fcc). Peaks of Al-enriched CoO decrease; however, the peak of diffuse scattering remains the same.

For the sample calcined at 500 °C, peaks of CoO and the most intensive peak of metallic Co (fcc) can be seen after reduction at T = 400 °C (Figure 6). It can be concluded that this CoO has a high content of Al because the peak positions correspond to the Al-enriched CoO observed for the sample calcined at T = 850 °C. Moreover, the peak of diffuse scattering on the left side of the (111) peak can also be observed. Reduction at 500 °C leads to an increase in the intensity of the peaks of metallic fcc cobalt. After reduction at 580 °C, the XRD patterns change towards decreasing CoO peaks and increasing Co^0^ peaks. However, the peak of diffuse scattering remains the same.

Structure refinement via the Rietveld method showed that the calculated XRD patterns of the CoO structure (rock salt structure, space group Fm3m) do not give good agreement with the experimental XRD patterns of the sample with the Al content x = 1/6 calcined at 500 °C and further reduced at T = 500 °C (Figure 7a) and T = 580 °C (Figure 7b). The intensity of the calculated 200 peak is smaller, and the calculated 111 peak is higher (and wider) than the experimental ones. There is also a discrepancy in the region of the diffuse scattering peak at ~38° 2θ. In some works [45,47], the structure of mixed Mg-Al oxide was analyzed as having some features of the spinel-like structure (Sp. gr. Fd3m). Such a consideration is based on the fact that both the rock salt and the spinel structures have the cubic close packing of O^2−^ anions. In the rock salt structure, there are four O^2−^ anions per unit cell that form four octahedral holes occupied by cations. At the same time, eight tetrahedral holes remain empty. In the spinel structure, there are 32 oxygen anions per unit cell (position 32e). They form 64 tetrahedral and 32 octahedral holes. Cations occupy 1/8 tetrahedral (8a) sites and 1/2 octahedral (16d) sites. The cubic spinel unit cell contains eight times more oxygen anions relative to the rock salt structure. Therefore, the unit cell volume and the lattice constant of the spinel structure are eight- and two-fold higher relative to the rock salt ones. In contrast to the space group Fm3m, for the space group Fd3m, the octahedral sites are not equivalent and are denoted as 16c and 16d. The transition from the Fm3m space group to the Fd3m one, where the spinel tetragonal 8a positions are partially occupied, gives the best agreement with the experimental data in the region of peaks (111) and (200) (Figure 7). In addition, better agreement is achieved in the region of the diffuse scattering peak. Additionally, the R factors decrease from 0.043 to 0.023 and from 0.051 to 0.026 for the samples reduced at T = 500 and 580 °C, respectively. It can be seen that the occupancy of spinel tetrahedral 8a positions increases with the calcination temperature (Table 2). At the same time, the occupancy of the rock salt 16c octahedral positions decreases. It can be assumed that an increase in reduction temperature makes Co^2+^ ions leave the rock salt octahedral positions and occupy the spinel tetrahedral ones. Thus, the structure becomes more and more spinel-like. Taking into account that the peaks of metallic cobalt became more intensive at T = 580 °C relative to T = 500 °C, the Co^2+^ content in mixed Co(II)-Al(III) oxide decreases with temperature due to the reduction of Co^2+^ to Co^0^. Therefore, it seems that the structure of CoO with a higher content of Al^3+^ ions is transformed towards the CoAl_2_O_4_ with an increase in reduction temperature. Such a structure can be considered as an intergrowth of two structures—CoO and CoAl_2_O_4_ (or Al_2_O_3_)—with a common oxygen sublattice. According to our XRD results, the CoO structure prevails, and according to the TPR data, stoichiometric CoAl_2_O_4_ is not formed. The formation of stoichiometric CoAl_2_O_4_ under these conditions can be excluded because its reduction requires relatively high temperatures of ~887 °C [48], as was shown on the standard sample, and there is no sign of that peak on the TPR profiles.

## 4. Discussion

The calcined samples are Co_3_O_4_-like and have the aforementioned cubic spinel structure. However, aluminum distribution differs depending on the calcination temperature. According to our TEM analysis, samples calcined at T = 850 °C consist of the particles with an Al-depleted interior with a structure close to the Co_3_O_4_ and Al-enriched surface. The reduction of such particles occurs as follows: at a temperature of ~345 ÷ 350, the reduction of Co^3+^ to Co^2+^ takes place in both the Al-depleted and the Al-enriched regions. The presence of an Al-enriched surface slows down the reduction of the interior’s Co^3+^ ions compared to pure Co_3_O_4_. However, the reduction of Co^2+^ to Co^0^ occurs at different temperatures. CoO with a low Al content is reduced at 430–440 °C, and CoO with a high Al content is reduced at 600–620 °C (Figure 1).

The reduction of the samples calcined at 500 °C is even more difficult than the reduction of the samples calcined at 850 °C. According to our TEM analysis, the main part of the particles is Al-enriched, and there are few Al-depleted particles. After reduction at 400 °C, the CoO-like phase prevails, and its quantity does not grow with further increases in temperature. Therefore, the first two TPR peaks at T = 300(305) and T = 345(350) °C can be attributed to the reduction of Co^3+^ to Co^2+^ in Al-depleted and Al-enriched spinel-like particles, respectively. The reduction peak at T = 405 °C can be assigned to Co^2+^→Co^0^ in Al-depleted CoO particles. The further reduction of Al-enriched CoO particles slows down. This mixed Co(II)-Al(III) oxide has features of both the rock salt and the spinel structures. Rietveld refinement of the structure in the frame of Fd3m space group showed that the mixed oxide contains cations in the spinel 8a tetrahedral positions and vacancies in the rock salt 16c octahedral ones that can be considered as an intergrowth CoO-CoAl_2_O_4_-type (or CoO-Al_2_O_3_-type) structure wherein CoO is predominant. Its reduction proceeds very slowly since the Al^3+^ ions that are in the immediate environment complicate the Co^2+^ reduction because Al^3+^ ions retain O^2−^ anions more strongly than Co^2+^ (standard electrode potentials are equal to −0.277 and −1.663 eV for Co^2+^/Co^0^ and Al^3+^/Al^0^, respectively).

During the reduction process, Al^3+^-enriched CoO is gradually transformed. With the increase in reduction temperature, the structure of mixed Co(II)-Al(III) oxide is additionally enriched by Al^3+^ ions and changes from a rock salt structure to a spinel one. Therefore, it becomes more CoAl_2_O_4_-like and consequently more difficult to reduce. However, stoichiometric CoAl_2_O_4_ is not formed. A comparison of the TPR profiles of the samples calcined at T = 500 and 850 °C showed that the samples calcined at lower temperatures contained an additional high-temperature peak at T = 690 °C. It seems that this peak can be assigned to the additional Al enrichment of Co(II) oxide during reduction at temperatures higher than 500 °C.

Our results can help to understand some results concerning the type of «strong interaction» in the alumina-supported [40] or promoted [49] cobalt catalysts and the influence of the interaction on their activity. It was shown that Co_3_O_4_/γ-Al_2_O_3_ catalysts synthesized via incipient wetness impregnation (IWI) and subsequent combustion synthesis (CS) contained well-dispersed Co species and abundant Co^3+^ ions, which played a crucial role in the catalytic combustion of CH_4_ [40]. However, the concentration of surface Co^3+^ ions on the catalysts prepared via the IWI method with subsequent calcination at 700 °C was relatively low, and the Co^3+^ was difficult to reduce to Co^2+^, which led to the lower activity of these catalysts. The average crystallite sizes of Co_3_O_4_ in the catalysts prepared via IWI with CS were smaller than in the catalysts prepared via IWI. It turned out that relatively large Co_3_O_4_ particles are more difficult to reduce than smaller ones. In general, a weaker interaction gives larger particles and higher reducibility, whereas a stronger interaction gives smaller particles and lower reducibility [50]. Earlier in this article, it was suggested that, in the IWI catalysts, the surface Co^2+^ species were stabilized by strong interactions with the alumina and that this stabilization led to a decreased Co^3+^/(Co^2 +^+ Co^3+^) ratio [40]. However, the type of this strong interaction remains unclear. We believe that the decreased content of surface Co^3+^ ions is associated with the enrichment of the Co_3_O_4_ surface with Al^3+^ ions, which substitute surface Co^3+^ ions, as in our coprecipitated samples calcined at 850 °C. This can explain the very complicated reduction of Co^3+^ to Co^2+^ in large Co_3_O_4_ particles. On the contrary, it seems that, in the IWI/CS, catalysts, the interaction of Co_3_O_4_ with alumina leads to the formation of (Co,Al)_3_O_4_ solid solution where Al^3+^ ions are distributed over the volume of Co_3_O_4_ particles, as in our coprecipitated samples calcined at 500 °C. This is indicated by the positions of the peaks on the TPR profiles for the two most active IWI/CS catalysts with Co_3_O_4_ loading of 30 and especially 50 wt%. The XRD patterns recorded ex situ show that the reduction-resistant CoO exists after reduction at 300 °C, and trace CoO remains even after reduction at 670 °C. Unfortunately, the asymmetry of the 111 CoO peak indicating the intergrowth of CoO and Al_2_O_3_ (clearly visible on the XRD patterns of the coprecipitated samples) cannot be seen on the XRD patterns of the IWI/CS catalysts due to the overlapping of the diffuse scattering peak and the broad 200 alumina peak (2θ~30–35, CuKα). In another work, Co_3_O_4_–Al_2_O_3_ catalysts were prepared through the aerosol-assisted self-assembly method with subsequent calcination at 450 °C [49]. High activity and stability were obtained simultaneously with the help of a small quantity of efficiently dispersed aluminum (10 at.% or x = 1/10). Through combining the results of H_2_–TPR and in situ XRD, it was found that the alumina interacts strongly with the cobalt species. More exactly, the authors wrote that it is the interaction between alumina and cobalt that restricts the further crystal growth of metallic cobalt species and prevents the agglomeration of catalysts during NH_3_ decomposition. It was shown by STEM-EDS that the Co and Al elements are mainly uniformly dispersed for the fresh catalysts (Co_3_O_4_-Al_2_O_3_) and still have a homogeneous dispersion for the used catalysts (metallic Co-Al_2_O_3_). However, it remains unclear whether Al^3+^ ions are uniformly distributed over the volume or surface. Based on our results, we can say that, in the fresh catalysts, the Al^3+^ ions are mainly uniformly distributed over the volume, forming (Co,Al)_3_O_4_ solid solution, which, upon the Co^3+^→Co^2+^ reduction, transforms into a mixed Co(II)-Al(III) oxide that, according to in situ XRD, exists in the range of T~300–600 °C. Despite the rather low signal/noise ratio due to cobalt fluorescence on the copper X-ray source, the asymmetry of the 111 peak was observed, which indicates the formation of the CoO-Al_2_O_3_ intergrowth structure. During the Co^2+^→Co^0^ reduction, phase segregation into metallic cobalt and Al_2_O_3_ takes place. Apparently, Al_2_O_3_, being located between the particles of metallic cobalt, prevents it from sintering.

## 5. Conclusions

The reduction of Co-Al mixed oxides with different distributions of Al^3+^ ions over particles was studied by using TPR and in situ XRD techniques. Mixed spinel-like oxides were prepared via coprecipitation with further calcination at 500 and 850 °C. We have shown that, in all cases, the reduction of Al-modified Co_3_O_4_ occurs via the following two-stage reduction: Co^3+^ →Co^2+^ →Co^0^. The addition of Al^3+^ ions decreases the average crystallite size of the Co(II)-Co(III) oxide and stabilizes the intermediate Co(II) oxide under reducing conditions by shifting the reduction towards a high-temperature region. Calcination at 500 °C leads to the formation of an inhomogeneous (Co,Al)_3_O_4_ solid solution, which is reduced to Al^3+^-containing CoO oxide at T < 400 °C. This mixed oxide has features of the rock salt structure seen in CoO and contains cations in a tetrahedral formation (ordered as in spinel structure). During the reduction of Co^2+^→Co^0^ at T > 500 °C, this oxide gradually loses cobalt, thereby additionally enriching itself with Al^3+^ ions, and is transformed from a rock salt structure to a spinel one by vacating rock salt 16c octahedral positions and filling spinel 8a tetrahedral ones. Therefore, it becomes more CoAl_2_O_4_-like and consequently more difficult to reduce. However, stoichiometric CoAl_2_O_4_ is not formed. An increase in the calcination temperature from 500 to 850 °C causes changes such as an increase in the average crystallite size of the initial oxide and segregation of Al^3+^ ions on the surface of particles. As a result, during the reduction of Co^3+^→Co^2+^, the formation of two types of intermediate CoO—Al-depleted and Al-enriched—can be detected. The reduction of CoO with low or high Al content differs in temperature and takes place at T ~ 430–440 °C and T = 600–620 °C, respectively.

## Figures and Tables

**Figure 1 materials-16-06216-f001:**
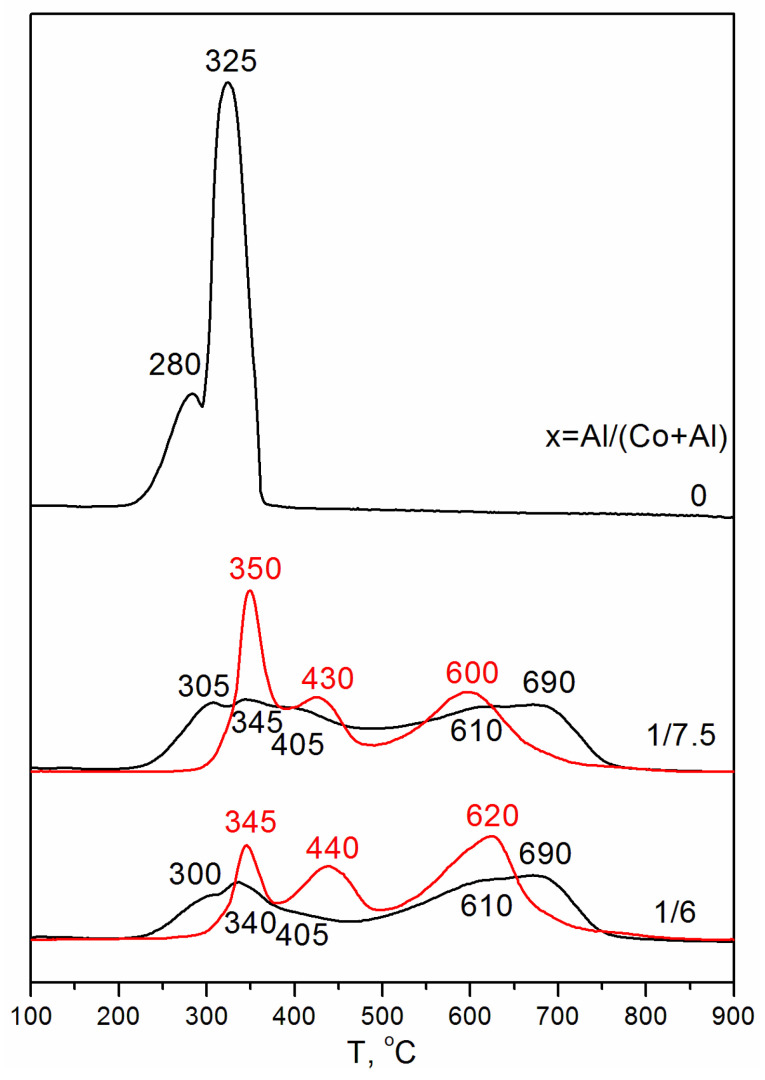
TPR profiles for samples with the Al mole fraction x = 0, 1/7.5 and 1/6 calcined at 500 °C (black lines) and for samples with x = 1/7.5 and 1/6 calcined at 850 °C (red lines). The profiles are normalized to the area under the peaks.

**Figure 2 materials-16-06216-f002:**
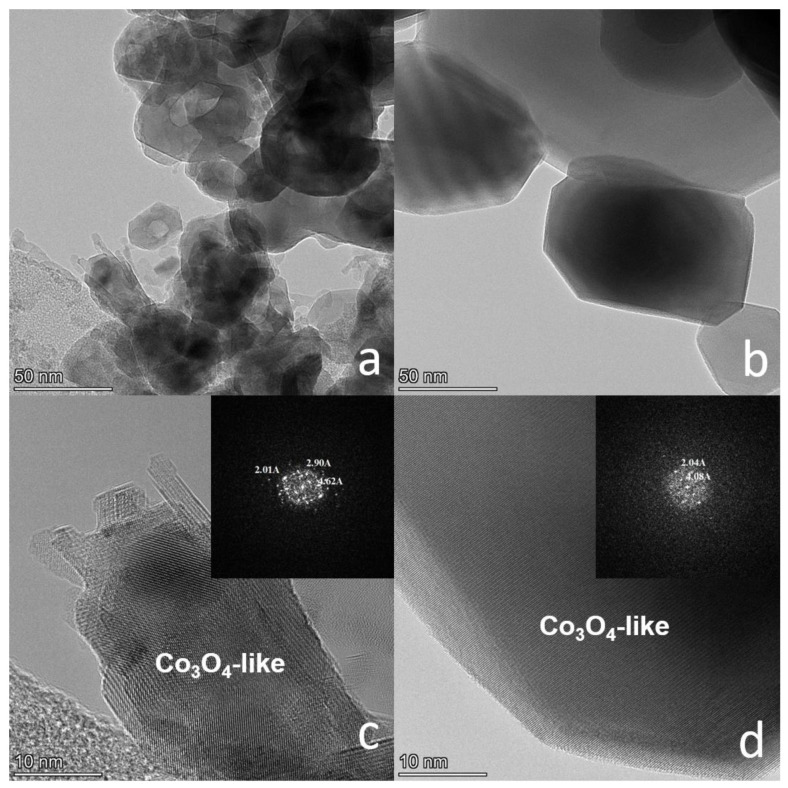
TEM images for the Co-Al sample with the Al fraction x = 1/6 calcined at (**a**) T = 500 °C and (**b**) T = 850 °C; HRTEM images combined with FFT for the Co-Al sample with the Al fraction x = 1/6 calcined at (**c**) T = 500 °C and (**d**) T = 850 °C.

**Figure 3 materials-16-06216-f003:**
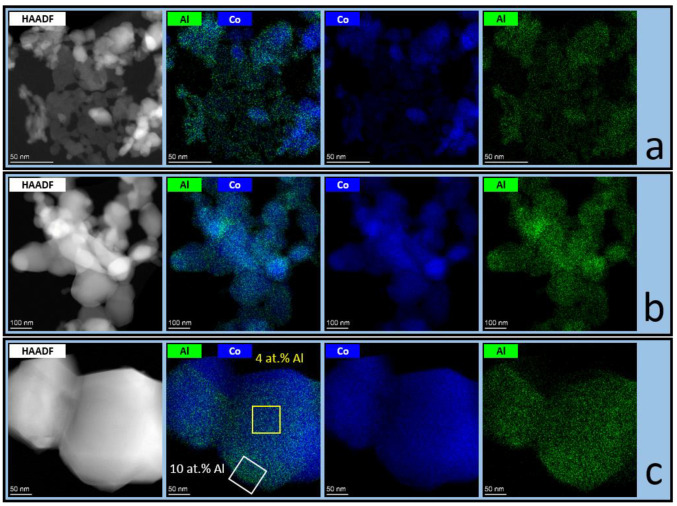
HAADF-STEM images combined with EDS mapping of Co (blue) and Al (green) atoms for Co-Al sample with the Al fraction x = 1/6 calcined at (**a**) T = 500 °C and (**b**,**c**) T = 850 °C.

**Figure 4 materials-16-06216-f004:**
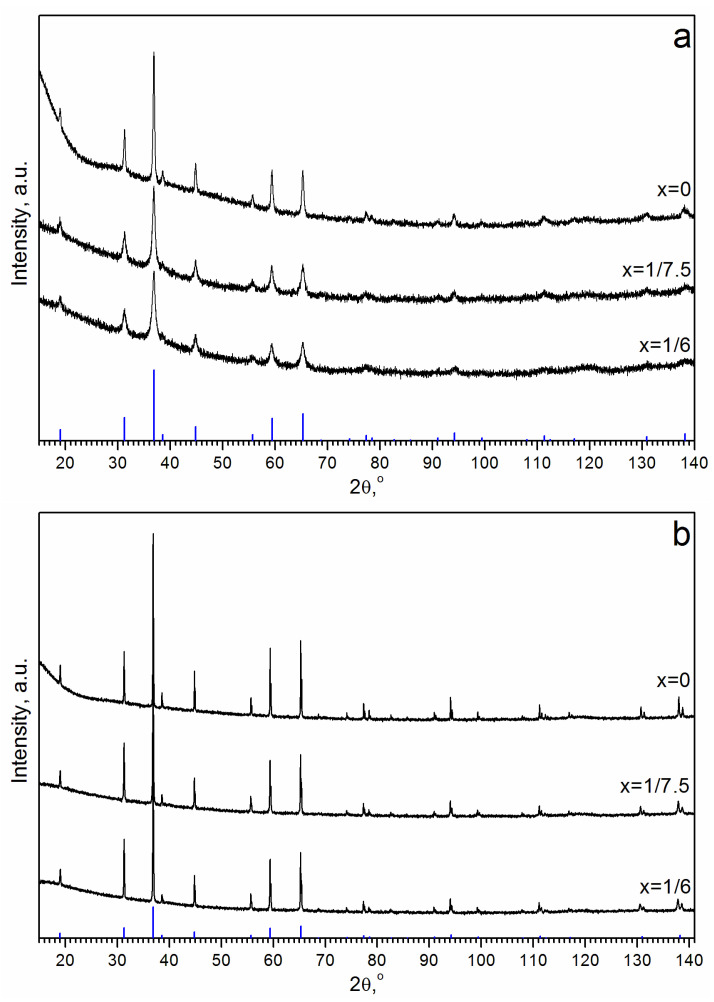
XRD patterns for the Co-Al oxide sample with the Al fraction x = 0, 1/7.5 and 1/6 calcined at (**a**) T = 500 °C and (**b**) T = 850 °C. The vertical bars (blue) at the bottom represent standard diffraction data from the PDF file for Co_3_O_4_ (PDF#43-1003).

**Figure 5 materials-16-06216-f005:**
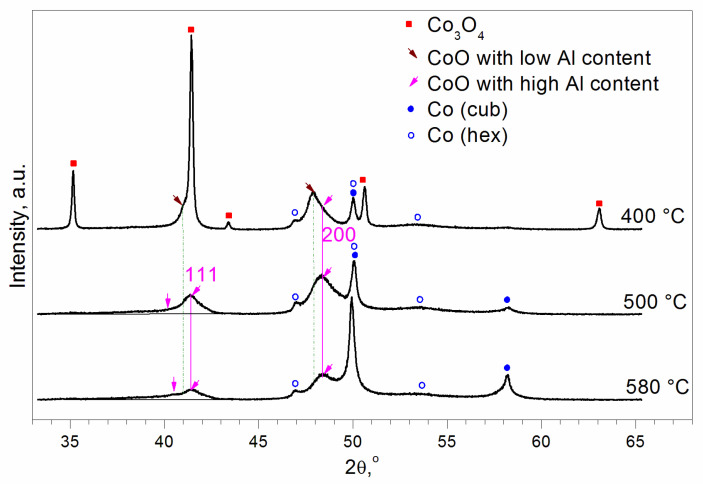
XRD patterns for the sample with the Al content x = 1/6 calcined at 850 °C and reduced in situ in a H_2_ flow at 400, 500, and 580 °C. The vertical arrow shows the peak of diffuse scattering, which is absent for CoO (Sp.gr. Fm3m, rock salt structure).

**Figure 6 materials-16-06216-f006:**
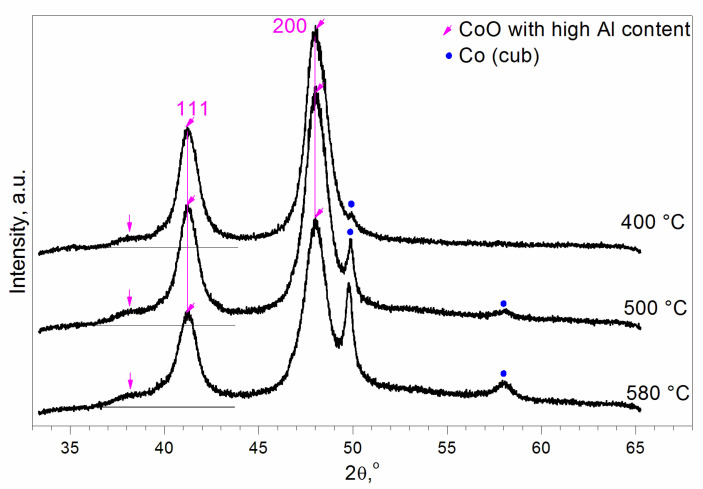
XRD patterns for the sample with the Al content x = 1/6 calcined at 500 °C and reduced in situ in a H_2_ flow at 400, 500, and 580 °C. The vertical arrow shows the peak of diffuse scattering, which is absent for CoO (Sp.gr. Fm3m, rock salt structure).

**Figure 7 materials-16-06216-f007:**
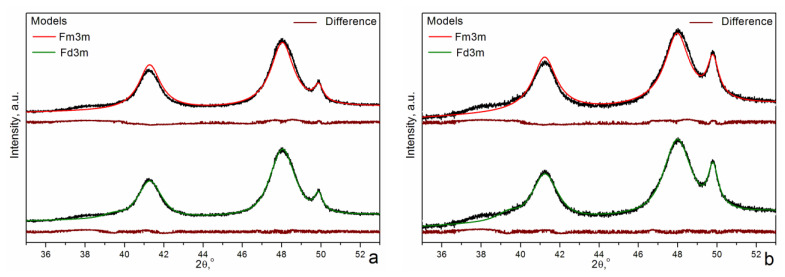
Structure refinement via the Rietveld method for the sample with the Al content x = 1/6 calcined at 500 °C and reduced at (**a**) T = 500 °C and (**b**) T = 580 °C.

**Table 1 materials-16-06216-t001:** Al^3+^-modified Co_3_O_4_: Rietveld refinement of lattice constants (a) and determination of average crystallite sizes (D) for the experimental XRD patterns of Co-Al samples with the Al fraction x = 0, 1/7.5 and 1/6 calcined at T = 500 and 850 °C (Figure 3).

T, °C	x = 0		x = 1/7.5		x = 1/6	
	a, Å	D, nm	a, Å	D, nm	a, Å	D, nm
500	8.085(1)	24	8.087(1)	12	8.091(1)	10
850	8.085(1)	135	8.088(1)	90	8.090(1)	90

**Table 2 materials-16-06216-t002:** Results of structure refinement via the Rietveld method in the frame of the Fd3m space group for the sample with the Al content x = 1/6 calcined at 500 °C and then reduced at T = 500 °C (Figure 7a) and T = 580 °C (Figure 7b).

T, °C	CoO (Fd3m)					Co	
	a, Å	D, nm	8a	16d	16c	a, Å	D, nm
500	8.513(1)	5.6	0.13	1.0	0.75	3.556(1)	13
580	8.513(1)	5.7	0.19	1.0	0.72	3.561(1)	12

a—lattice constant; D—average crystallite size; 8a and 16d—occupancies of spinel tetrahedral and octahedral positions, respectively; 16d and 16c—occupancies of rock salt octahedral positions in space group Fd3m.

## Data Availability

No new data were created or analyzed in this study. Data sharing is not applicable to this article.

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
