# Peer review of "Reducibility of Al3+-Modified Co3O4: Influence of Aluminum Distribution"

_materials, 2023, doi:10.3390/ma16186216_

Round 1

Reviewer 1 Report

This manuscript could be considered for publication after addressing the following comments and questions.
1.     The introduction section lacks a description of the background and significance of the study.

2.     The Co-based catalysts are described cursorily in the Introduction section, please provide more information on Co-based catalysts.
3.       In some reactions, the synergistic effect is reached with use of CoO-Co composite catalystsandIt worth noting that Co3O4 is easily reduced to metallic cobalt via two sequential steps: Co3O4→CoO→Co.,Please find references to support these statements.
4.     With regard to all similar misrepresentations in the article, such as "270 ÷ 415 °C" and "6% Co ÷ 20%Co" ect, please replace "÷" to "-".

5.     For the modification of Co oxides by Al3+ ions in the introduction, please give 2-3 more examples.

6.     Please provide more details about the experimental preparation, such as information of the raw materials and chemical manufacturers.

7.     In the H2-TPR profiles, why is there no significant difference between the H2-TPR curves of 1/7.5 and 1/6 Co-Al samples calcined at 500°C? Please give an explanation.

8.     It seems to be, that high temperature induces migration of Al3+ ions to the particle surface, and they can be considered as core–shell particles, which have an Al-depleted core and an Al-enriched shell (Figures 2b and 2c). Please find references to support this statement.

9.     Where is the XRD pattern for the 3.3 XRD section? Please provide the XRD pattern for part 3.3 and list the information such as lattice parameters for samples at different calcination temperatures in the table..

10.  “Peaks of CoO are shifted to the higher diffraction angles that indicates a decrease in the lattice constant with temperature.”, “After reduction at T = 400 °C, two types of CoO containing relatively low and relatively high concentration of Al3+ ions are formed. This is manifested by the asymmetry of CoO 200 diffraction peak at 2θ = 47–50°.” and “Moreover, for charge balance, cationic sublattice in Al-en-riched CoO should contain vacancies, which also decrease the lattice constant.” Please find references to support the three statements.

11.  Actually, it can be seen in Fig. 4 that the CoO peak of the sample calcined at 500 °C showed no significant change with the increase of reduction temperature, please explain the reason.

12.    Table 1 is not clearly labeled, making it challenging to understand the data presented. Please provide the necessary information on the essential parameters at the bottom of the Table.

The English language should be carefully revised.

Reviewer 2 Report

In this work, the authors studied the reducibility of Al-modified cobalt oxide. The work is interesting, however, it lacks originality, as similar studies have already been published. Furthermore, the paper is relatively short since it lacks catalytic experiments. The following comments should be considered:

1.The main issue studied here is the reducibility of Al-modified Co3O4. It has been studied through a temperature-programmed reduction. The paper lacks the originality, as the TPR studies of similar materials have already been published. See, for example, https://doi.org/10.1007/s11426-018-9261-5 or http://dx.doi.org/10.1016/j.apcatb.2014.12.016. You must explain exactly what is new in your study and place your results in context with previous investigations.  

2.You studied only a very narrow composition range (Al/(Al+Co) ratio between 0-0.17). You must explain how you chose these compositions and why.

3.The major applications of the materials are found in the catalysis field (see the Introduction). However, no catalytic studies are presented in the manuscript. It is a major handicap. The lack of catalysis studies reduces the value of the paper considerably. I would recommend resubmitting your paper as a short communication instead of full paper. Otherwise, the catalysis studies must be included.

4.The STEM images in Fig. 2. have poor resolution. You need to provide high resolution TEM images. The selected area diffraction patterns should also be included in the paper to confirm the phases.

5.The authors claim that they prepared core-shell particles with an Al-depleted core and Al-enriched shell (lines 16, 205, line 310, etc.). However, they based their claim on EDS results only (see Fig. 2). You need to confirm this via HR TEM with an atomic resolution. Otherwise, you cannot claim it.

Round 2

Reviewer 1 Report

The revised manuscript could be accepted now due to excellent revision.

Reviewer 2 Report

Authors answered my comments. The paper is acceptable for publication.